# Phenolic Contents, Organic Acids, and the Antioxidant and Bio Activity of Wild Medicinal Berberis Plants- as Sustainable Sources of Functional Food

**DOI:** 10.3390/molecules27082497

**Published:** 2022-04-12

**Authors:** Liu Yang, Zhenyuan Zhang, Xiaoying Hu, Lixin You, Raja Asad Ali Khan, Yan Yu

**Affiliations:** 1College of Food Science and Engineering, Changchun University, Changchun 130022, China; yangliu811981@sina.com (L.Y.); zzyzzy0817@sina.com (Z.Z.); huxiaoying0608@sina.com (X.H.); 2College of Life Science, Changchun Sci-Tech University, Changchun 130600, China; yuyan_i0608@sina.com; 3Department of Plant Pathology, The University of Agriculture Peshawar, Peshawar 25130, Pakistan; 4Key Laboratory of Green Prevention and Control of Tropical Diseases and Pests, Ministry of Education (School of Plant Protection), Hainan University, Haikou 570228, China

**Keywords:** phytochemicals, functional food, antibacterial, food borne bacteria

## Abstract

Wild fruits have increasingly been investigated as part of recent searches for food products with a high antioxidant activity. In this study, wild edible berberis *Berberis vulgaris* collected from three different provinces (Jilin, Heilongjiang, and Liaoning) were investigated for their phenolic contents, organic acid contents, mineral contents, antioxidant activity as well as their antimicrobial potential against a range of common food borne pathogens. In addition, a physiochemical and mineral analysis of the fruits was also performed. The methanol extracts of berberis fruit collected from Jilin province were highly active against all the studied food borne bacterial pathogens, i.e., *S. aureus* and *L. monocytogenes*, *E. coli*, *P. fluorescens*, *V. parahaemolyticus*, and *A. caviae* while the berberis extracts from Heilongjiang and Liaoning showed activity only against Gram-negative bacteria. The phenolic content and antioxidant activity were determined by the HPLC separation method and β-carotene bleaching methods, respectively. Four organic acids such as malic acid, citric acid, tartaric acid, and succinic acid were identified while a variety of phenolic compounds were detected among which catechin, chlorogenic acid, and gallic acid were found to be the predominant phenolic compounds in all three of berberis fruit samples. The berberis fruit from Jilin was found to be superior to the Heilongjiang and Liaoning fruit regarding desired physiochemical analysis; however, there were no significant differences in the mineral contents among the three samples. Overall, the berberis fruit from Jilin was ranked as the best in term of the nutritional, physiochemical, antimicrobial, and antioxidant properties. This study confirms the various useful characteristics and features of berberis at a molecular level that can be used as a sustainable source for their potential nutritional applications for making functional foods in different food industries.

## 1. Introduction

Various species of the genus *Berberis* which occur around the world are cultivated and grown for specific purposes [1]. *Berberis vulgaris*, a prominent member of the genus *Berberis* in the Berberidaceae family, is native to northwest Africa, western Asia, and southern and central Europe. It grows at altitudes of 800–1500 m in a variety of arid and rocky soils, though it is primarily cultivated in cooler regions [2,3,4]. On some occasions, the fruits are found individually, while in others they appear in the form of a bunch of grapes [2]. The yellow bunch of flowers that blossom in April or May consist of 15–25 flowers. The fruit has a 3–5 mm width and 7–10 mm length and is an oblong red berry [5]. The fruit is mostly used for the extraction of juice in medication and food industries or can also be cooked with rice or other dishes [6,7]. The *B. vulgaris* fruit comprises several nutrients including malic acid, dextrose, tartaric acid, fructose, citric acid, resin, and pectin. It is also rich in iron, potassium, calcium, and vitamins A and C [8,9]. The fruits of *B. vulgaris* have been reported to contain zinc, manganese, iron, copper, 2% protein, 16.24% carbohydrate, 79.6% humidity, 0.99% ash, and 1.16% fat [7,10]. It is also extensively used as a food additive [5]. The medicinal history of the fruit can be traced back to 2500 years ago in several countries while in Chinese medicine it was used 3000 years ago [3,4]. 

In modern and traditional medicine, different parts of *B. vulgaris* such as the fruit, roots, and bark are the commonly investigated parts of this plant [11]. An alkaloid berberine is the predominant bioactive compound of *B. vulgaris*, which has been known to produce several anti-inflammatory, antimicrobial, antidiabetic, and antioxidant properties [11]. Extracts obtained from fruit have many useful characteristics that are beneficial to the nervous system and cardiovascular system, so have potential uses in treating certain neuronal diseases such as contraction, epilepsy, and hypertension [12,13]. The fruits of *B. vulgaris* contain high levels of vitamin C, organic acids, anthocyanins, and tannins [2]. The strong antifungal activity and antibacterial activity of the fruit extracts have also been documented by researchers [14]. 

As the popularity of *B. vulgaris* increases globally, evidence has shown that this plant has unlimited reservoirs of compounds that are useful both in medicine and human nutrition. Because of the structural diversity in their active ingredients, they are becoming useful sources for new therapeutics. There is a growing demand to explore the sources of various useful compounds that can be used to produce functional foods with good nutrition and medical uses. Nutrient-rich wild fruits such as *B. vulgaris* are very suitable for the pharmaceutical and food industries [15]. Due to scientific research on the medicinal value, nutritional composition, and advantageous features of the wild edible fruits of *B. vulgaris* grown all over the world, the fruit has aroused great interest [16]. These fruits are also important for antioxidant potential [17]. The type and amount of phytochemical contents varies considerably among various sources as well as in different geographical regions. For example, catechins, a phenolic compound, varies according to its source when collected from different sites [18]. Keeping in view the availability of diverse phytochemical constituents, in this study the fruits of *B. vulgaris* collected from three different geographical regions, i.e., Jilin, Heilongjiang, and Liaoning provinces were evaluated for their nutritionally important phytochemical constituents such as organic acids and phenolic compounds. The study was also extended to investigate the antioxidant activity and antimicrobial effect of fruit extracts against common food borne bacterial pathogens.

## 2. Results

### 2.1. Organic Acid Contents of B. vulgaris Fruits

The organic acid contents of *B. vulgaris* fruits collected from three regions were determined (Table 1 and Figure 1). A total of four organic acids, i.e., citric, tartaric, malic, and succinic acids were found with different concentrations. The highest concentrations of citric acid (1.798 g kg^−1^) were seen in the Jilin samples followed by Liaoning (1.468 g kg^−1^) and Heilongjiang (0.873 g kg^−1^). Tartaric acid was significantly higher in the Heilongjiang fruits (1.134 g kg^−1^) while in Jilin and Liaoning tartaric acid was in almost similar concentrations, i.e., 0.901 g kg^−1^ and 0.911 g kg^−1^, respectively. Malic acid was in similar concentrations in the Jilin samples (1.585 g kg^−1^) and Liaoning (1.339 g kg^−1^) while significantly lower amount of malic acid was noticed in the Heilongjiang samples (1.108 g kg^−1^). Succinic acid was also significantly higher in Jilin followed by Heilongjiang and Liaoning. 

### 2.2. Phenolic Content and Antioxidant Activity

Different phenolic compounds were detected in *B. vulgaris* fruits with different concentrations (Table 2 and Figure 2). Generally, Jilin’s samples contained significantly higher concentrations of most of the phenolic compounds as compared to Heilongjiang and Liaoning that had almost similar concentrations. The highest concentrations of gallic acid, catechin, chlorogenic acid, vanillic, caffeic, syringic, ferulic acid, rutin, and o-coumaric in Jilin’s fruits were recorded as 0.182 g kg^−1^, 0.640 g kg^−1^, 0.624 g kg^−1^, 0.044 g kg^−1^, 0.089 g kg^−1^, 0.049 g kg^−1^, 0.031 g kg^−1^, 0.073 g kg^−1^ and 0.051 g kg^−1^, respectively. The two compounds p-coumaric and quercetin were present in similar concentrations in all three samples. It was noticed that compared to other compounds, catechin, chlorogenic acid, and gallic acid were found to be the predominant phenolic compounds in all three fruits. The antioxidant property of *B. vulgaris* fruits was determined by the β-carotene bleaching method (Figure 3). The results indicated significantly higher antioxidant levels in Jilin’s fruits that gave 92.34% antioxidant activity followed by Jilin, Heilongjiang 68.23%, and Liaoning 71.45%. The Heilongjiang and Liaoning samples exhibited statistically similar antioxidant activities.

### 2.3. Physiochemical Properties

The data regarding the physiochemical properties of *B. vulgaris* fruits are presented in Table 3. The values for all the physiochemical properties were significantly higher in Jilin’s samples compared to Heilongjiang and Liaoning. Although minor differences were noticed between Heilongjiang and Liaoning except moisture, crude cellulose, and pH, they were not significantly different from each other. The highest dry matter (32.28%), water-soluble DM (27.57%), ash (0.93%), aw (0.97), pH (5.58), moisture (88.37%), reduced sugar (6.14%), crude protein (13.67%), crude cellulose (10.36%), and crude oil (0.81%) were recorded for Jilin’s samples. 

### 2.4. Mineral Contents

The mineral contents of fruits were calculated and are presented in Table 4. The data show that all the studied minerals were present in all the three types of samples. However, it was noted that some key minerals, i.e., Ca, Fe, K, Mg, and Na were present in significantly higher concentrations in Jilin’s samples than in the Heilongjiang and Liaoning samples. The concentrations of Ca, Fe, K, Mg, and Na were noted as 2868.56 ppm, 403.57 ppm, 13,113.31 ppm, 1363.16 ppm, and 2725.24 ppm, respectively in the fruits collected from Jilin, Heilongjiang, and Liaoning. Notably, compared to the other minerals, all the samples predominantly contained Ca, Fe, K, Mg, Na, and P while mineral contents such as Bi, Co, Li, and V were found in lower amounts.

### 2.5. Antimicrobial Activity

Fruit extracts were tested for their antibacterial potential against a range of food borne bacterial pathogens through the well diffusion method. Among the three fruit extracts, the samples collected from Jilin showed the highest antibacterial activity followed by Heilongjiang and Liaoning (Figure 4, Table 5). Interestingly, the extracts from Jilin’s fruits inhibited the in vitro growth of all the studied bacteria, i.e., *S. aureus* and *L. monocytogenes*, *E. coli*, *P. fluorescens*, *V. parahaemolyticus*, and *A. caviae*. The zone of inhibition produced by the Jilin fruit extracts were 9.1 mm, 10.2 mm, 14.1 mm, 15.3 mm, 14.0 mm, and 16.2 mm against *S. aureus* and *L. monocytogenes*, *E. coli*, *P. fluorescens*, *V. parahaemolyticus*, and *A. caviae*, respectively. The extracts from the Heilongjiang fruits were only active against *E. coli*, *P. fluorescens*, *V. parahaemolyticus*, and *A. caviae* giving 5.0 mm, 7.0 mm, 6.1 mm, and 6.2 mm zones of inhibition, respectively. The Liaoning fruit extracts inhibited the growth of *P. fluorescens*, *V. parahaemolyticus*, and *A. caviae* and exhibited 5.0 mm, 7.1 mm, and 6.0 mm zones of inhibition, respectively. Generally, the extracts were more active against Gram-negative bacteria (*E. coli*, *P. fluorescens*, *V. parahaemolyticus*, and *A. caviae*) than Gram-positive bacteria (*S. aureus* and *L. monocytogenes*). The positive control streptomycin gave the maximum inhibition zone while the negative control methanol showed no activity against all the bacteria.

## 3. Discussion

In this study, citric acid, tartaric acid, and malic acid were found to be the predominant organic acids in *B. vulgaris* fruits particularly in those collected from Jilin province. Citric acid is listed as an ingredient in a significant percentage of prepared foods, beverages, and medications. Its main use is as preservative and a flavoring agent and in food and beverages [19]. Tartaric acid has a stronger, sharper taste than citric acid. Although it is renowned for its natural occurrence in grapes, it also occurs in apples, cherries, papayas, peaches, pears, pineapples, strawberries, mangoes, and citrus fruits. Tartaric acid is used preferentially in foods containing cranberries or grapes, notably wines, jellies, and confectioneries [20]. Malic acid, a universal acidulant, is usually related to apples but is also served as the main acidic component of lingonberries, cranberries, guavas, and grapes. Tartaric acid is useful in masking the unpleasant flavors of food and gives a tart taste, lowers the pH, and has antimicrobial effects [20]. Organic acids are determinants of various physiological food characteristics such as maturity, formation, and taste and also play a role in human nutrition and health [21,22]. The sour or sweet food taste also depends on the type of organic acid and its level in food [23]. 

Phenolic acids are the most commonly phenols found in foods. In addition to their nutritional advantages, they also affect the antioxidant and sensory properties of food. Phenolic compounds are abundant in a stable daily diet which includes adequate amounts of whole grains, fruits, and vegetables [24]. In our studies, phenolic compounds such as catechin, chlorogenic acid, and gallic acid were found to be the predominant phenolic compounds. Catechin, a well-proven antioxidant phytochemical compound, is an active pharmacological agent in terms of lipoprotein oxidation inhibition, the inhibition of the Kruppel-like factor expression [25,26], showing enhanced antifungal activity of amphotericin B on *Candida albicans* [27], and preventing human plasma oxidation [28]. Chlorogenic acid is a compound found in a wide variety of foods and beverages, including fruits, vegetables, olive oil, spices, wine, and coffee. Chlorogenic acid and related hydrolysates act as antioxidants. They also have liver protective activity, can inhibit cancer, and can also help to treat chronic HBV infection [29]. Gallic acid (GA), a member of the hydroxybenzoic acids, represents a big family of secondary metabolites and is found widely in the plant kingdom. It also acts as a natural antioxidant [30]. Generally, GA can prevent the oxidative damage of biomolecules by ROS [31,32]. Other useful properties such as anticancer, antimicrobial, antifungal, and anti-inflammatory have also been reported for GA [33,34]. Our results are consistent with several previous studies in which different levels of the phenolic content in berberis fruits were reported [11,35,36,37]. The determination of antioxidant activity is one of the reasons for measuring phenolic contents or the techno-functional characteristics of the wild fruits. In this study, the berberis fruits collected from Jilin showed a significantly higher antioxidant activity than the Heilongjiang and Liaoning fruits. The potent antioxidant compounds are responsible for the antioxidant activity. For example, antioxidant action of catechin has been well-established by various in vitro, in vivo, and physical methods [38]. Chlorogenic acid exhibited potent antioxidant activity by increasing superoxide dismutase, catalase, and glutathione and reducing lipid peroxidation in rats [39]. The higher antioxidant activity of Jilin’s fruits can be explained on the basis of more content of antioxidant compounds such as catechin, chlorogenic acid, and gallic acid in Jilin’s fruits than in the Heilongjiang and Liaoning fruits. The results of our study are in line with previous findings reported in different studies. The ethanol and water extracts of *B. vulgaris* fruits were previously reported to have 73.62% and 82.52% antioxidant activity, respectively [37]. In another study, the antioxidant activity of *B. vulgaris* fruits were shown as 75.01–90.64% [35]. 

The results for the physico-chemical profiles of the fruits highlighted significant differences. Jilin’s fruits exhibited a significantly higher pH value than the Heilongjiang and Liaoning fruits. A higher pH value is desirable as it maintains the shelf life of the food product. Researchers have highlighted that a high pH is important to maintain the sensory characteristics, i.e., the taste and flavor during a product’s shelf life [40]. Significantly higher values for dry matter were seen in Jilin’s samples than the Heilongjiang and Liaoning samples. A similar trend was also observed for ash and protein. The results regarding the physico-chemical profile of berberis fruits observed in this study were consistent withother studies. Previous reports have shown dry matter contents of 31.22% and 32.77%, ash contents of 0.65% and 3.44% for *B. vulgaris* and *B. crataegina*, respectively [41,42]. The pH range of 4.17–5.58 noted in our study is similar to 5.5 and 3.35 already reported for berberis fruits [40,42]. Similarly, the aw value for berberis fruits obtained in this study is parallel to the aw values of 0.95 and 0.94 reported previously for *B. vulgaris* and 0.94 in *B. crataegina* [42]. 

Results regarding the antimicrobial potential of berberis fruits extracts revealed their antibacterial activity against common food borne pathogenic bacteria. However, the antibacterial activity of the berberis fruits extracts showed lower antibacterial activity than the Streptomycin control. For some strains such as *S. aureus* and *L. monocytogenes* and *E. coli*, their inhibition was zero for B2 and B3 treatments. The antibacterial effect could be attributed to biologically active phenolic compounds present in berberis fruits extracts. Catechin, a predominant compound in this study was previously reported to have a strong antibacterial activity because of the generation of its hydrogen peroxide [43,44]. Similarly, Chlorogenic acid and caffeic acid have also been reported as a strong antibacterial compounds. In a study, chlorogenic acid showed a potent antibacterial effect against Gram-positive bacteria *B. subtilis*, *S. aureus*, *S. pneumonia*, and Gram-negative bacteria *E. coli*, *Shigella dysenteriae* and *Salmonella typhimurium* [45]. The antibacterial activity could also be attributed to some minerals having antibacterial properties such as Mg, Zn, and Co. In previous studies, a significant growth inhibition effectiveness of zinc and Mg against bacteria even at lower concentrations was reported [46,47]. Similarly, berberine, a phytochemical present in berberis, was also reported to have strong antibacterial properties [48]. The extracts of berberis fruits collected from Jilin province were more active in terms of antibacterial activity then the fruits collected from other provinces. Except *S. aureus*, all other studied bacteria were highly sensitive to the extracts of berberis fruits collected from Jilin province. This might be due to the fact the significantly higher antimicrobial phenolic contents (catechin, chlorogenic acid, caffeic acid) and minerals (Mg, Zn, Co) in the berberis fruits collected from Jilin province. The results of this study are in line with the already reported results where a strong antimicrobial effect of *B. psedumbellat*, *B. calliobotrys*, and *B. orthobotrys* fruits was reported [49]. It was noticed that the Gram-positive bacteria were more resistant to the fruit extracts. This is because the cell walls of Gram-positive bacteria have a thick peptidoglycan layer, which is more resistant than the thin layer of peptidoglycan in Gram-negative bacteria [48].

The overall comparison showed that the berberis fruits collected from Jilin province were superior to the samples collected from Heilongjiang and Liaoning provinces regarding the nutritional, physiochemical, and antimicrobial analysis. Jilin’s fruits contained higher levels of useful organic acids, phenolic compounds, and mineral contents and showed more antioxidant and antimicrobial activity. There were significant variations among the three fruits collected from the three different regions. These variations might have been due to the genetic structure and environmental conditions, i.e., soil conditions, light, humidity, and temperature [49,50]. Recently, in another study, it was suggested that the phytochemical constituents could be affected by geographical conditions or growing conditions of the fruits [51]. The findings of this study suggested that the berberis fruits collected from Jilin province were better than the berberis fruits collected from other two provinces regarding their nutritional, physiochemical, and antimicrobial characteristics.

## 4. Materials and Methods

### 4.1. Berberis vulgaris Fruits Collection

Plants of *B. vulgaris* in equal size and age were collected from three provinces, i.e., Jilin, Heilongjiang, and Liaoning in June 2020 and authenticated by a botanist. Equal size and age of *B. vulgaris* fruits were separated, rinsed with tap water, and stored at −80 °C for further analysis.

### 4.2. Organic Acid Determination

For the extraction of organic acids, a previously reported method was used with little modification [52]. The samples were fragmented at 500 g and from each sample 25 g was taken into the centrifuge tube. The samples were homogenized after adding 0.009 N 10 mL of H_2_SO_4_ and mixed for one hour with a shaker at 80 RPM. The samples were than centrifuged for 20 min at 15,000× *g*. The supernatant was filtered through coarse filter paper first and then through 0.45 μm filter twice and last in the SEP-PAK C18 cartridge. Organic acid concentrations were evaluated by using HPLC with Aminex column (1100 series HPLC G 1322 A, Agilent Technologies, Waldbronn, Germany). Organic acids were detected at 214 and 280 nm wavelengths. As the mobile phase, 0.009 N H_2_SO_4_ were passed through 0.45 μm filter membrane. Standards included in the analysis were Citric acid, Tartaric acid, Malic acid, Succinic acid. Analyses were performed five times.

### 4.3. Phenolic Compounds Determination

The HPLC separation method was used for the determination of phenolic compounds [53]. The samples were fragmented at 500 g and from each sample 25 g was taken into the centrifuge tube. The samples were centrifuged for 15 min at 15,000× *g* after mixing homogenously and 1:1 dilution with distilled water. The supernatant after passing through 0.45 μm membrane filter was injected into HPLC gradient system. The chromatographic separation as performed in HPLC 1100 series using DAD detector with 4 μm ODS column of 250 mm × 4.6 mm. Spectral measurements were performed at 254, and 280 nm. The methanol: acetic acid: water (10:2:88) and methanol: acetic acid: water (90:2:8) were used as mobile phase solvent A and B respectively with 20_l injection volume and flow rate of 1 mL/min. Calibration curves were generated using Gallic acid, Catechin, Chlorogenic acid, Vanillic, Caffeic, Syringic, P-coumaric, Ferulic acid, O-coumaric, Quercetin, 4-hydroxybenzoic acid. Analyses were completed five times.

### 4.4. Physicochemical Analysis

For the analysis of dry matter contents, 3 g samples were dried at 105 °C. The percent dry matter contents were obtained by comparing dry weight/fresh weight of the samples. Water soluble dry matter was determined by using Abbe refractometer (Model Ra 250HE, Kyoto Electronics Manufacturing Ltd., Tokyo, Japan). For ash calculation, samples were process in ash oven until the appearance of light gray-white color. After which the samples were weighed, and the percentage of ash content was determined. The pH of the crushed fruit juice was calculated by using a pH meter (6173 brand, Jenco, Shanghai, China). The homogenized fruit samples were checked for water activity (aw) by using water activity device (Aqua Lab brand, Decagon, Inc., Washington, DC, USA). The methods reported by [54,55] were used for measuring crude cellulose, protein, oil, reducing sugar and moisture contents. Analyses were completed five times.

### 4.5. Mineral Analysis

For the determination of mineral contents, 15 mL of pure HNO_3_ was added to the 0.5 g of fruit sample (grounded and dried). The samples were incinerated at 200 °C in microwave oven (MARS 5 Plus) dilution was performed with sterilize water. Mineral concentrations were measured with ICP-AES (inductively coupled plasma-atomic emission spectrometry) [56]. The processing conditions were set as plasma gas flow rate = (axial) 15 L/min and (radial) 10.5 to 15.0 L/min, auxiliary gas flow rate = 1.5 L/m, viewing height = 5 to 12 mm, copy and reading time and 1 to 5 s, copy time = 3 s and RF power = 0.7 to 1.5 kW. Standards were run to generate the calibration curve. Analyses were completed five times.

### 4.6. Antioxidant Evaluation

For determining antioxidant activity, first the fruits were extracted by previously reported process with little modifications [57]. Briefly, the fruits were dried for three days at 55 °C and crushed slightly. A total of 30 mL of sterilize water was added to 3 g of dried crushed fruits and solution was shaken for 15 h. The samples were transferred to centrifuge tubes and centrifuged (2800× *g*) for 15 min at 4 °C. After centrifugation, the samples were passed to 110 mm filter papers extract obtained was stored at −20 °C till further use. The antioxidant activity of the fruit extract was evaluated through β-carotene bleaching method as described previously. Butylated hydroxyanisole (BHA) was used as the standard. The degradation rate was calculated according to first-order kinetics using the equation DR = In(a/b) × 1/t (“ln” is natural log, “a” is the initial absorbance (470 nm) at time 0, “b” is the absorbance (470 nm) at 100 min, and “t” is time). The antioxidant activity (AA), was expressed as percent of inhibition relative to the control, using the formula AA = (DR control − DR sample)/DR control × 100. Analyses were completed five times.

### 4.7. Antimicrobial Evaluation

The methanol extract of fruits (5 g/200 mL) was evaluated for their antibacterial potential against Staphylococcus aureus, Listeria monocytogenes, Escherichia coli, Pseudomonas fluorescens, Vibrio parahaemolyticus and Aeromonas caviae through agar well diffusion method. At first, each plate was poured with 25 mL NA (nutrient agar) medium, having 0.5 mL bacterial suspension (10^8^ cfu mL^−1^), and allowed to cool. With the help of 2 mm diameter borer total of five wells (5 mm) were punched in each plate. Three wells were poured with 10 µL of fruit extract one well with 10 µL of standard antibiotic as a positive control and one with 10 µL of methanol as a negative control. The plates were incubated at 28 °C for 24 h and zone of inhibition was measured with the help of transparent plastic ruler. Each treatment was replicated four times and means values were presented after statistical analysis. Analyses were completed five times.

### 4.8. Statistical Analysis

Experiments were performed using five replicates and data were presented as mean ± SD. SPSS software (version 21.0) (SPSS, Inc., Chicago, IL, USA) was used for statistical analysis. Analysis of variance (ANOVA) was applied to analyze the data and means were separated and compared using LSD test. Differences were determined to be significant at *p* < 0.05.

## 5. Conclusions

It is concluded that fruits of *B. vulgaris* collected from three provinces, i.e., Jilin, Heilongjiang, and Liaoning showed different concentrations of nutritional, physiochemical, and mineral contents. The fruits also differed in terms of antioxidant and antimicrobial activities. The results suggest that different geographical locations significantly affect the phytochemical constituents of *B. vulgaris* fruits. The fruits collected from Jilin province exhibited more contents of the desired phytochemical constituents and showed a higher antioxidant and antimicrobial activity than the Heilongjiang and Liaoning fruits.

## Figures and Tables

**Figure 1 molecules-27-02497-f001:**
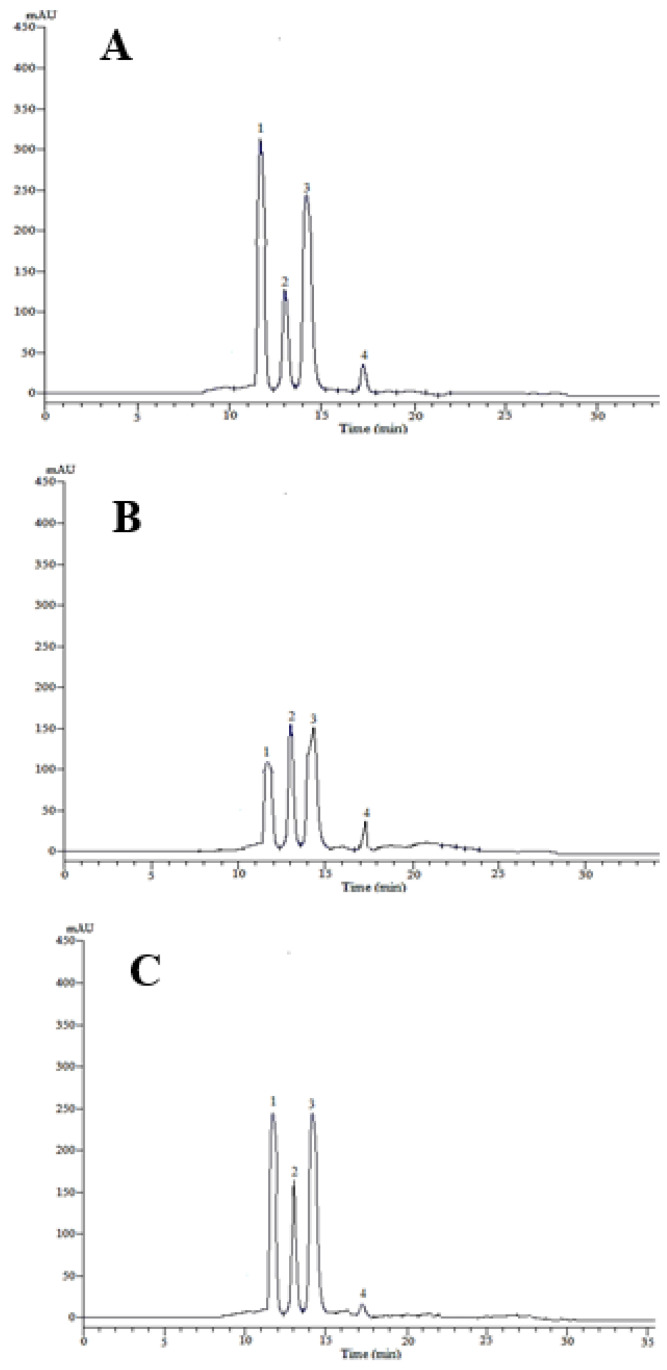
HPLC chromatograms of organic acids detected in *Berberis vulgaris* fruits collected from (**A**): Jilin, (**B**): Heilongjiang and (**C**): Liaoning province. 1: citric acid, 2: tartaric acid, 3: malic acid and 4: succinic acid.

**Figure 2 molecules-27-02497-f002:**
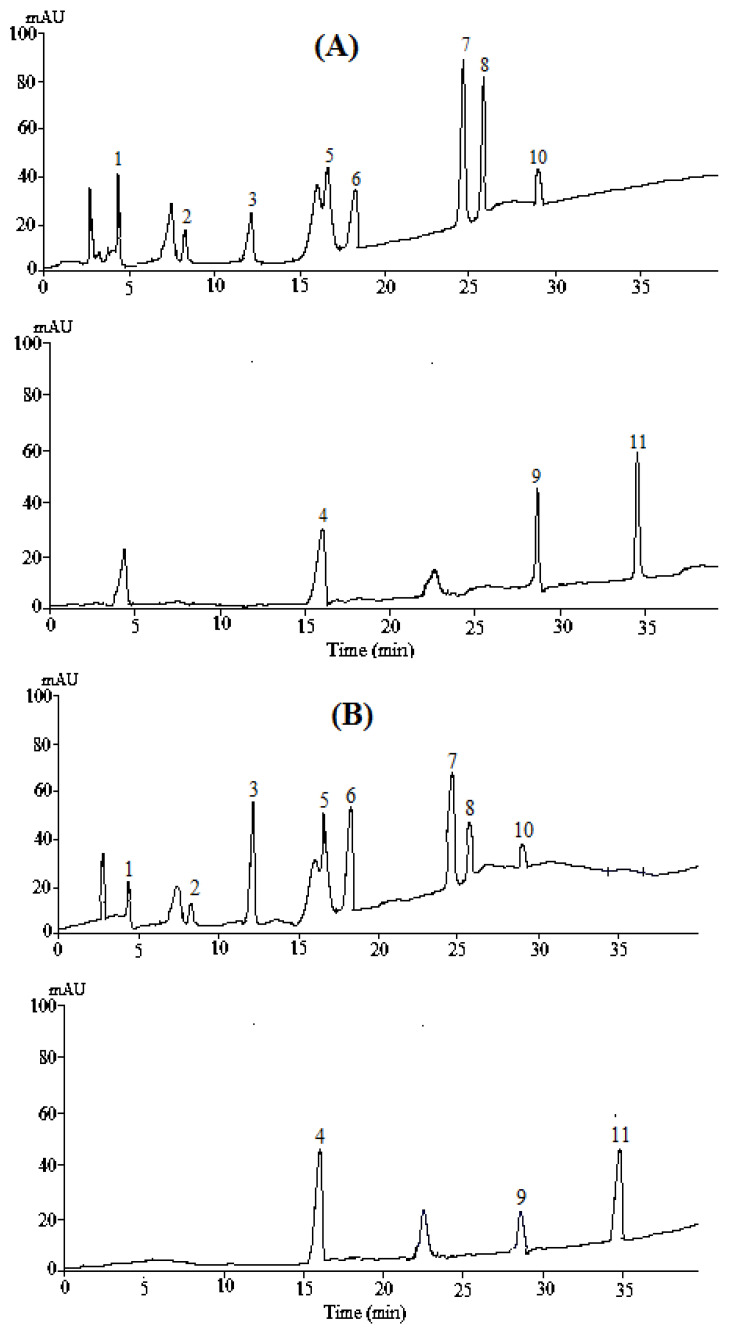
Phenolic contents detected in *Berberis vulgaris* fruits collected from (**A**): Jilin province, (**B**): Heilongjiang province and (**C**): Liaoning province. 1: Gallic acid, 2: Catechin acid, 3: Chlorogenic acid, 4: Vanillic acid, 5: Caffeic acid, 6: Syringic acid, 7: P-coumaric 8: Feulic, 9: Rutin acid, 10: O-coumaric acid and 11: Quercetin acid.

**Figure 3 molecules-27-02497-f003:**
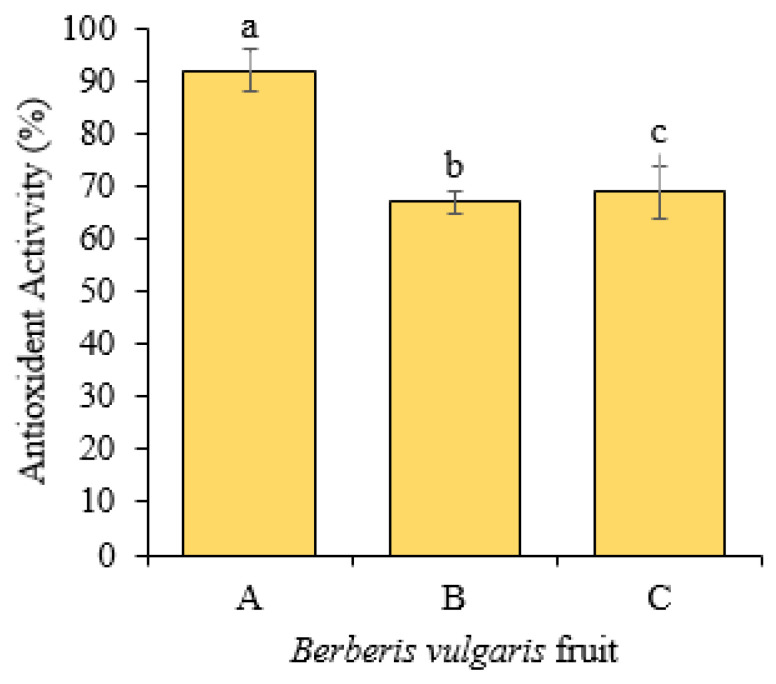
Antioxidant activity (%) of *Berberis vulgaris* fruits collected from A: Jilin province, B: Heilongjiang province and C: Liaoning province. Each value is a mean of five replicates ± SD. Similar lettering shows no significant difference according to LSD test.

**Figure 4 molecules-27-02497-f004:**
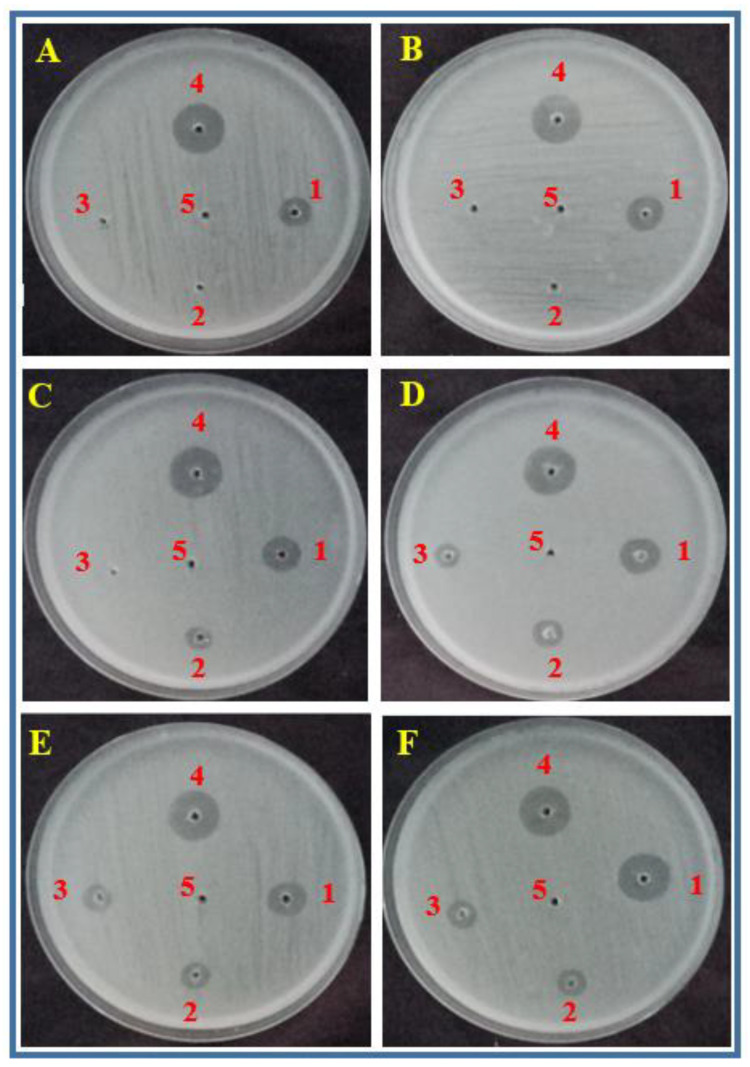
Zones of inhibition produced by *Berberis vulgaris* fruit extracts. 1: Jilin province, 2: Heilongjiang province, 3: Liaoning province, 4: Positive control (antibiotic) and 5: negative control (methanol) against food born bacteria (**A**): *S. aureus*, (**B**): *L. monocytogenes*, (**C**): *E. coli*, *(***D**): *P. fluorescens*, (**E**): *V. parahaemolyticus* and (**F**): *A. caviae*.

**Table 1 molecules-27-02497-t001:** Organic acids detected in berberis fruits.

Organic Acids	Content (g/kg)		
	B1	B2	B3
Citric acid (g/kg)	1.798 ± 0.037 a	0.837 ± 0.018 c	1.468 ± 0.028 b
Tartaric acid (g/kg)	0.901 ± 0.018 b	1.134 ± 0.045 a	0.911 ± 0.023 b
Malic acid (g/kg)	1.585 ± 0.028 a	1.108 ± 0.068 c	1.339 ± 0.012 b
Succinic acid (g/kg)	0.126 ± 0.004 a	0.116 ± 0.007 b	0.101 ± 0.006 c

Each value is a mean of five replicates ± SD. Different lettering in a row indicates significant differences among the fruits according to LSD test. *Berberis vulgaris* fruits collected from B1: Jilin province, B2: Heilongjiang province and B3: Liaoning province.

**Table 2 molecules-27-02497-t002:** Phenolic compounds detected in berberis fruits.

Phenolic Contents	Content (g kg^−1^)		
	B1	B2	B3
Gallic acid	0.182 ± 0.023 a	0.137 ± 0.016 b	0.142 ± 0.018 b
Catechin	0.640 ± 0.012 a	0.590 ± 0.021 b	0.601 ± 0.021 b
Chlorogenic acid	0.624 ± 0.024 a	0.455 ± 0.018 c	0.475 ± 0.022 b
Vanillic	0.044 ± 0.001 a	0.026 ± 0.001 b	0.029 ± 0.001 b
Caffeic	0.089 ± 0.001 a	0.062 ± 0.002 b	0.065 ± 0.001 b
Syringic	0.049 ± 0.001 a	0.037 ± 0.001 b	0.036 ± 0.001 b
P-coumaric	0.029 ± 0.003 a	0.027 ± 0.002 a	0.029 ± 0.001 a
Ferulic acid	0.031 ± 0.001 a	0.023 ± 0.003 b	0.021 ± 0.002 b
Rutin	0.073 ± 0.001 a	0.059 ± 0.004 b	0.055 ± 0.004 b
O-coumaric	0.051 ± 0.002 a	0.034 ± 0.002 b	0.035 ± 0.001 b
Quercetin	0.021 ± 0.001 a	0.022 ± 0.001 a	0.021 ± 0.001 a

Each value is a mean of five replicates ± SD. Different lettering in a row indicates significant differences among the fruits according to LSD test. *Berberis vulgaris* fruits collected from B1: Jilin province, B2: Heilongjiang province and B3: Liaoning province.

**Table 3 molecules-27-02497-t003:** Physiochemical profiles of berberis fruits.

Parameters	Contents		
	B1	B2	B3
Dry matter (DM) %	32.28 ± 2.3 a	22.67 ± 1.1 b	21.71 ± 1.4 b
Water-soluble DM (%)	27.57 ± 1.8 a	21.78 ± 0.8 b	21.65 ± 1.3 b
Ash (%)	0.93 ± 0.07 a	0.63 ± 0.02 b	0.69 ± 0.04 b
Aw	0.97 ± 0.5 a	0.81 ± 0.1 b	0.83 ± 0.03 b
pH	5.58 ± 1.0 a	4.12 ± 0.6 b	2.17 ± 0.1 c
Moisture (%)	88.37 ± 4.7 a	58.67 ± 3.8 b	48.25 ± 2.3 c
Reducing sugar (%)	6.14 ± 0.8 a	5.87 ± 0.4 b	4.82 ± 0.3 c
Crude protein (%)	13.67 ± 1.2 a	8.61 ± 0.8 b	8.92 ± 0.9 b
Crude cellulose (%)	10.36 ± 0.9 a	8.72 ± 1.1 b	6.82 ± 0.3 c
Crude oil (%)	0.81 ± 0.05 a	0.56 ± 0.03 b	0.58 ± 0.03 b

Each value is a mean of five replicates ± SD. Different lettering in a row indicates significant differences among the fruits according to LSD test. *Berberis vulgaris* fruits collected from B1: Jilin province, B2: Heilongjiang province and B3: Liaoning province.

**Table 4 molecules-27-02497-t004:** Mineral contents detected in berberis fruits.

Minerals	Mineral Contents (ppm)		
	B1	B2	B3
Ag	48.84 ± 3.3 a	46.21 ± 2.3 a	45.26 ± 3.7 a
Al	19.11 ± 1.2 a	17.68 ± 1.1 a	17.24 ± 1.1 a
As	4.67 ± 0.3 a	4.64 ± 0.5 a	4.34 ± 0.2 a
B	68.47 ± 4.2 a	63.23 ± 3.2 a	65.28 ± 4.3 a
Ba	1.61 ± 0.1 a	1.58 ± 0.04 a	1.63 ± 0.5 a
Bi	0.28 ± 0.03 a	0.26 ± 0.01 a	0.27 ± 0.01 a
Ca	2868.56 ± 58.3 a	2439.02 ± 13.2 b	2454.35 ± 17.2 b
Co	0.32 ± 0.02 a	0.31 ± 0.02 a	0.32 ± 0.01 a
Cr	37.89 ± 2.4 a	37.13 ± 3,6 a	35.47 ± 4.2 a
Cu	5.12 ± 0.5 a	4.85 ± 0.4 a	4.95 ± 0.3 a
Fe	403.57 ± 4.8 a	320.36 ± 8.4 b	302.48 ± 9.2 c
K	13,113.31 ± 723.0 a	11,216.26 ± 578.2 b	11,176.25 ± 845.4 b
Li	0.41 ± 0.01 a	0.43 ± 0.01 a	0.39 ± 0.1 a
Mg	1363.16 ± 6.3 a	1123.10 ± 5.8 b	1098.47 ± 32.5 b
Mn	6.47 ± 0.8 a	6.60 ± 0.04 a	6.50 ± 0.3 a
Na	2725.24 ± 8.3 a	2498.12 ± 13.6 b	2396.45 ± 12.5 c
Ni	22.58 ± 1.2 a	21.24 ± 1.4 a	21.46 ± 0.8 a
P	2568.37 ± 12.2 a	2536.43 ± 16.8 a	2576.43 ± 15.3 a
Sr	18.14 ± 1.2 a	19.46 ± 2.1 a	19.26 ± 1.4 a
V	0.82 ± 0.04 a	0.88 ± 0.2 a	0.83 ± 0.3 a
Zn	9.47 ± 1.4 a	8.97 ± 1.1 b	9.23 ± 0.8 b

Each value is a mean of five replicates ± SD. Different lettering in a row indicates significant differences among the fruits according to LSD test. *Berberis vulgaris* fruits collected from B1: Jilin province, B2: Heilongjiang province and B3: Liaoning province.

**Table 5 molecules-27-02497-t005:** Antimicrobial activity of berberis fruits against common food borne pathogenic bacteria.

Bacteria	Zone of Inhibition (mm)				
	Treatments			Control	
	B1	B2	B3	Streptomycin	SDW
*S. aureus*	9.1 ± 1.1 b (S)	0 ± 0.0 c (NS)	0 ± 0.0 c (NS)	18.0 ± 0.5 a (VS)	0 ± 0.0 c (NS)
*L. monocytogenes*	10.2 ± 0.8 b (VS)	0 ± 0.0 c (NS)	0 ± 0.0 c (NS)	18.0 ± 0.7 a (VS)	0 ± 0.0 c (NS)
*E. coli*	14.1 ± 1.6 b (VS)	5.0 ± 0.6 c (NS)	0 ± 0.0 d (NS)	18.4 ± 0.4 a (VS)	0 ± 0.0 d (NS)
*P. fluorescens*	15.3 ± 1.3 b (VS)	7.0 ± 0.5 c (S)	5.0 ± 0.8 c (NS)	17.8 ± 0.6 a (VS)	0 ± 0.0 d (NS)
*V. parahaemolyticus*	14.0 ± 1.5 b (VS)	6.1 ± 1.1 c (S)	7.1 ± 1.0 c (S)	18.1 ± 0.5 a (VS)	0 ± 0.0 d (NS)
*A. caviae*	16.2 ± 1.2 a (VS)	6.2 ± 0.8 b (S)	6.0 ± 0.5 b (S)	18.0 ± 0.5 a (VS)	0 ± 0.0 c (NS)

Each value is a mean of five replicates ± SD. Different lettering in a row indicates significant differences among the treatments according to LSD test. *Berberis vulgaris* fruits collected from B1: Jilin province, B2: Heilongjiang province and B3: Liaoning province. SDW: Sterilized Distilled Water. NS = No sensitive (ZI: 0–5 mm); S = Sensitive (ZI: 5.1–10); VS = Very sensitive (ZI: higher than 10 mm).

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
