# Peer review of "Phenolic Contents, Organic Acids, and the Antioxidant and Bio Activity of Wild Medicinal Berberis Plants- as Sustainable Sources of Functional Food"

_molecules, 2022, doi:10.3390/molecules27082497_

Round 1

Reviewer 1 Report

None

Author Response

Thank you for accepting our article and recommending for publication in Molecules Journal. 

Reviewer 2 Report

rows 281-283 which findings in your study suggest that "berberis fruits could be used as sufficent phytochemical sources for the developpment of functional foods"? pay attention to formulate proper conclusions

row 300- it is neccesarly to show also the standards chromatograms

row 311- the standards chromatograms must be presented also

Author Response

Thank you again for providing valuable suggestions. Your careful observations and indicated comments improved the quality of our manuscript a lot. 

Comment: rows 281-283 which findings in your study suggest that "berberis fruits could be used as sufficient phytochemical sources for the development of functional foods"? pay attention to formulate proper conclusions.

Answer: Yes, this statement was not correct and our study is not providing sufficient findings for this information. Thank you for the suggestions. We now deleted this statement in our revised version and restated as “The findings of this study suggested that berberis fruits collected from Jilin province are better than berberis fruits collected from other two provinces regarding their nutritional, physiochemical and antimicrobial characteristics. We also delete the similar statements from the conclusion section. 

Comment: row 300- it is neccesarly to show also the standards chromatograms

row 311- the standards chromatograms must be presented also

Answer: Yes, you are right. These chromatograms are necessarily to show. Actually, most of the papers we cited and published in previous literature did not contain chromatograms for standards. That’s why we also did not carefully save chromatograms for standards as part of our study. We used them just to analyze our data. This study was conducted in June, 2020. We tried our best to find chromatograms of standards in our saved data but not succeeded. Anyhow, still if you think these chromatograms are very necessary, we will run analysis for standards again and put chromatograms in the manuscript. But, as these days we are not working in labs so it will be difficult for us to do this. Therefore, we are requesting you to give us a relax in this matter. Hopefully, we will get relaxation as a lot of other studies also did not included standards chromatograms in the final manuscript.

This manuscript is a resubmission of an earlier submission. The following is a list of the peer review reports and author responses from that submission.

Round 1

Reviewer 1 Report

the methods mentioned in the paper are uncompletely described, whitout calibration, validation and, when necessary, accurate description (e.g. the method used for antioxidant activity assessment).

it seems unprobably that the fruits of the same species with similar chemical content have different antibacterial activities. which of the molecules could influence the antibacterial activity?

there are no correlations between antioxidant activity, chemical content and antimicrobial activity.

the conclusions are not suitable for a scientific paper. 

the references are old and not at the point

Reviewer 2 Report

Reviewer comments

Introduction

P1, L42: The juice cure cholecytitis, is it a drug?

  1. Materials and Methods

P2, L83-86: Indicate, what were the conditions used for the determination of organic acid by HPLC (eluent, flow rate and standard used)? was realized calibration curve (R2)? What were the characteristics of the detector used?

P3, L131: How was realized methanol extract of fruits (proportions o conditions)?

P3, L134: What is NA medium? Explain, why Mueller Hinton Agar wasn't used?

P3, L135-136: What was the depth of the wells?

P3, L139: Please, Indicate the incubation conditions and the instrument used to measure the inhibition halos.

  1. Results

P5, L168: Please, to define SDW in the table 1.

P5, L168: Explain (Table 1), why is there significant differences of zero (0) vs zero (0) when ANOVA was realized? Additionally, is correct to compare Zero (0) inhibition with presence of inhibition?

P6, L183: Why the units of the organic acid in column 1 of the table 2 appear in g/kg and the table header (content) the text as g kg-1?

P10, L221: Add capital letter (antioxidant activity) and italic (Beberis vulgaris)

P11, L240-243: The authors indicate that there isn’t significant difference in the mineral contents but in the second sentence indicate that some minerals are present in significant concentrations, how explain this? Additionally, why are Ca, Fe, K, Mg and Na considered key minerals?

  1. Discussion

P13, L313-314: The antibacterial activity of the berberis fruits extracts showed lowest antibacterial activity than Streptomycin control, inclusive some strains as S. aureus and L. monocytogenes and E. coli, their inhibition was zero for B2 and B3 treatments. The authors should explain, why do you assume that the antibacterial activity of the berberis fruits extracts as strong?

Moreover, I suggest that a classification by the diameter of the inhibition halos (example: no sensitive, sensitive, very sensitive..) due be used.

P13, L315: The authors assumed that the antibacterial activity is only influenced by phenolic compounds present in berberis fruits extracts but some minerals as copper, zinc, and others, have showed antibacterial activity. Similarly, some organic acids and the berberine have showed also antibacterial activity. This topic must be improved and better discussed.

General Comments

*Please revise the English language.

*Please homogenize in all manuscript the abbreviature of genus and specie of Berberis vulgaris.

*Please use italic in all the genus and species of bacteria (P3, L166 and P5,  L132)

*To revise all manuscript and correct notations chemical and binomial nomenclature inadequately used (example: H2SO4, HNO3…).

Reviewer 3 Report

Authors need to improve the introduction

the results were presented in the opposite way as in the methodology!

LSD significant need to recheck in all data

the discussions of phenolic acids and antioxidant are weak and need more
in line 296 author wrote "in a study" and in line 298 wrote "in another study" that is not correct from my point of view